# Corporate Social Responsibility of Forestry Companies in China: An Analysis of Contents, Levels, Strategies, and Determinants

**Yanli Li [1],\* and Lan Gao [2]**

1   College of Forestry and Landscape Architecture, South China Agricultural University, 483 Wushan Road, Guangzhou 510642, China

2   College of Economics and Management, South China Agricultural University, 483 Wushan Road, Guangzhou 510642, China

\*   Correspondence: leeyanli@scau.edu.cn; Tel.: +86-0208-258-0256

**Abstract:** Corporate social responsibility (CSR) has gained attention in the forestry sector, especially among Chinese forestry companies, which faces serious challenges. This study explores the CSR activities of Chinese forestry companies listed on the Shanghai and Shenzhen Stock Exchanges and analyzes how they differ from their international counterparts. CSR contents, levels, strategies, and determinants are examined through a quantitative content analysis and statistical analyses. The results show that Chinese forestry companies' CSR contents, like their international counterparts, are diverse and include the environment, employees, communities, general social issues, consumers and products, investors and creditors, governments, and supply chains. Both of them focus on environmental and employee responsibility and pay less attention to community responsibility; however, their CSR priority activities differ. While Chinese companies rank employee responsibility activities first and environmental activities second, their international counterparts prioritize environmental activities over employee responsibility. Chinese forestry companies have four types of CSR strategies—reactive, focused, opportunistic, and proactive—and the majority of these companies adopt reactive strategies. Only a few Chinese forestry companies choose proactive strategies. Forest resources partially explain the variance in the levels of government responsibility among forestry companies, and the industry type influences the levels of corporate environmental responsibility.

**Keywords:** corporate social responsibility (CSR); Chinese forestry companies; CSR content; CSR level; CSR strategy; determinants

## 1. Introduction

Forestry faces such issues as ecological safety, climate change, and energy shortages and plays an important role in sustainable development. Forestry companies have become the targets of public criticism for their unsustainable use of forest resources, huge energy consumption, high exhaust emissions, and water pollution [1–4]. With globalization and increasing societal expectations about the sustainable use of forests, corporate social responsibility (CSR) practices in the forest sector have gained attention and become increasingly crucial because of the environmentally sensitive nature of forest-based businesses [5–7]. CSR can help forestry companies improve relationships with stakeholders, maintain legitimacy [8–10], address challenges [11], realize sustainable development [11,12], and achieve competitive advantages [6,11,13–16]. In China's transitioning economy, forestry companies face serious challenges associated with environmental deterioration, trade frictions, and irresponsible corporate behavior. For example, Chinese forestry companies are accused of using large amounts of illegal timber. With the Chinese government's increase in environmental protection efforts and the upgrading of

green trade barriers, CSR has become a new threshold for Chinese forestry companies to survive in the domestic and international markets. In this context, it is essential to study the CSR activities of Chinese forestry companies to determine the status of CSR as well as possible countermeasures.

Several studies discussed the motivations, contents, and influencing factors of CSR in forestry companies [6,17–23]. The review by Li and Gao [9] showed that forestry companies sampled in most studies are headquartered in North America or Europe, especially large forestry companies [9]. Further, little research has focused on forestry companies in emerging economies, including China. Most Chinese forestry companies are relatively small, which makes them different from their international counterparts. Additionally, CSR activities vary according to the contexts of forestry companies, including social and cultural backgrounds and institutional arrangements [5,6,21]. China's economic transition and institutional differences may cause the motivations, contents, strategies, and determinants of CSR in Chinese forestry companies to differ from those of their international counterparts. There is a lack of emphasis on these aspects in the literature.

This study seeks to bridge the knowledge gap by evaluating the CSR activities of Chinese forestry companies in the following manner. First, this study analyzes the contents, levels, and priorities of CSR in these companies and compares these activities with those of their international counterparts. Second, it analyzes the CSR strategies in Chinese forestry companies. Finally, it verifies the impact of some contextual factors on CSR levels and strategies. This study found that Chinese forestry companies implement diverse CSR activities, focus on environmental and employee responsibility, and pay less attention to community responsibility. Additionally, the majority of these companies adopt reactive CSR strategies. Finally, forest resources and industry type partially explain the variance in the levels of CSR for the government and environment, respectively.

The rest of this article is organized as follows. The second section introduces the theoretical background. The third section describes the research method and data resources. The fourth section presents the results. The fifth section discusses the results. The final section concludes the study.

## 2. Literature Review and Hypotheses

### 2.1. Contents and Levels of Corporate Social Responsibility

Corporate efforts toward CSR have intensified. CSR benefits companies by satisfying the expectations of society. From the stakeholder theory and resource-based view perspectives, the managers' decisions on CSR could maximize the utility of different stakeholders, improve the relationships with the primary or more influential stakeholders who control critical resources [18], and develop both the organizations' and stakeholders' dynamic capabilities, resources, and routines [24]. In addition, the adoption of CSR can create new knowledge which under some conditions could play a crucial role in developing resources and capabilities [25] for companies to build sustainable competitive advantages [16] according to the resource-based view, knowledge-based view, and intellectual capital-based approach. For example, they may attract better employees or increase current employees' motivation, morale, commitment, and loyalty to those companies [25].

It is necessary to distinguish between two divergent opinions on the definition of CSR. One is that CSR is the opposite of economic responsibility and voluntary beyond economic and legal requirements [26,27]. Another view is that economic responsibility and compliance with the law are dimensions of CSR. This study adopts the latter broader definition of CSR, which refers to the obligations of businesses to pursue sustainable policies, make sustainable decisions, and conduct activities that conform to the objectives and values of society [28]. However, the constituents of CSR are often biased toward specific interests. Carroll [29] proposed the classic pyramid model of CSR, which consists of economic, legal, ethical, and philanthropical responsibilities. Of these responsibilities, economic performance forms the foundation of CSR. Elkington [30] insisted that CSR must include not only traditional financial responsibility, but also environmental and social responsibilities. Environmental

and social responsibility are non-economic responsibility types. Jamali [31] classifies CSR into voluntary and mandatory CSR.

These studies demonstrated two CSR characteristics of forestry companies. First, CSR across forestry companies is diverse. These companies attempt to balance social, economic, and environmental objectives through CSR [6,32] and implement various perspectives of CSR under intensifying globalization and public pressure [23,32]. The CSR contents of forestry companies include employees, health, resource recycling, communities, energy, safety, culture, education, air, water, procurement, transportation, electricity, consumption, accountability, philanthropy, and indigenous residents [6,17,33–35]. The other characteristic of forestry companies' CSR is the prioritization of environmental responsibility over social responsibility, such as community involvement and human rights [36,37]. This prioritization can be attributed to the fact that forestry has a direct and deep influence on the natural environment. Environmental CSR activities rank first in these companies [6]. Forest certification is considered to be one of the most effective methods of verifying legality and sustainable practices [38].

## 2.2. Corporate Social Responsibility Strategies

The choice of CSR contents is a direct reflection of CSR strategies. The CSR strategy refers to how companies allocate resources to achieve their economic, social, and environmental goals. It represents corporate attitudes toward stakeholders. Forestry companies with different strategies prioritize CSR contents differently and transfer different quantities of resources to their chosen CSR categories.

In the literature, the classification of CSR strategies is mainly based on negative or positive attitudes. Galbreath [39] divided CSR strategies into shareholder, altruistic, reciprocal, and citizenship types, from the most reactive to the most proactive. Porter and Kramer [40] classified CSR strategies into two categories according to the fit between a CSR strategy and a corporate strategy: Responsive CSR and strategic CSR. Lee and Rhee [41] classified environmental strategies into four types—reactive, focused, opportunistic, and proactive—according to the depth and width of CSR. Companies with a reactive strategy are engaged in relatively few categories of CSR contents, with the lowest levels of CSR for each category. Companies with a focused environmental strategy usually prioritize a limited number of environmental categories but maintain higher CSR levels for these environmental issues. Companies with an opportunistic environmental strategy are engaged in many CSR categories, which are conducted at relatively average levels. Companies with a proactive strategy implement the highest levels of CSR in every category.

This study adopts Lee and Rhee's [41] definition of a CSR strategy, that is, a CSR strategy is the corporate selection of the width and depth of CSR activities. The width refers to the range of CSR activities and the depth refers to the levels of CSR activities. In this study, the range of CSR activities includes the environment, employees, communities, general social issues, consumers and products, governments, investors and creditors, supply chains, and other CSR stakeholder activities. Companies may choose different categories of concern and indulge at different levels among these eight categories.

## 2.3. Determinants of Corporate Social Responsibility

### 2.3.1. Company Size and Corporate Social Responsibility

Company size is one of the most important determinants of CSR in the forestry sector. Most studies in this area have shown the influence of company size on the contents and levels of CSR. Han and Hansen [35] found a significant positive relationship between company sales and the level of CSR implementation in forestry companies. Companies with higher annual sales tend to have a higher level of CSR implementation, while those with lower annual sales tend to have a lower level of CSR implementation. Vidal and Kozak [38] insisted that CSR contents have different focuses in different sized forestry companies. Large forestry companies incorporate all CSR activities, while small forestry companies focus on sustainable forest management, forest certification, and compliance.

According to resource-based theory, large forestry companies have more access to resources and greater resource slack, which have been found to significantly affect CSR commitment [14,42,43]. By contrast, small forestry companies often have constrained or inadequate resources, which may make it unviable for them to engage in CSR initiatives. Large forestry companies may also have better developed administrative processes, perceive and address the external environment differently given their business exposure. Thus, they may implement a broader range and higher levels of CSR than small forestry companies [42].

From the firm visibility and stakeholder theory perspectives, large forestry companies have a wider range of stakeholders and make a greater social impact, tend to be more visible, and are more likely to attract the public's attention. Therefore, such companies have more stakeholders and so are likely to be more socially responsible. Small forestry companies, on the contrary, may face fewer pressures or gain little recognition from CSR given their comparatively lower visibility. Therefore, large forestry companies are more motivated to implement CSR to achieve their social commitments, and thus their range of CSR initiatives is more extensive. In addition, when scale economies exist, the average costs of CSR are lower in large forestry companies [44]. This implies that such companies may provide more CSR attributes [45].

In summary, large forestry companies are more likely to engage in CSR behavior. Therefore, the contents and levels of CSR may vary among forestry companies of different sizes.

**Hypothesis 1a (H1a).** *Forestry companies of different sizes exhibit different CSR levels.*

**Hypothesis 1b (H1b).** *Forestry companies of different sizes implement different CSR strategies.*

2.3.2. Industry and Corporate Social Responsibility

While CSR is relevant across industries, some industries have a greater impact on the environment and/or society [5,46]. Societal expectations regarding social responsibility may vary from one industry to another [5,47]. There are no consistent conclusions on the effect of industry types on CSR. Waddock and Graves [48] found a great difference in CSR disclosure across industries. Simpson et al. [49] argued that there are great differences between industries with regard to CSR. While Cowen et al. [50] and Balabanis et al. [51] reported that CSR does not differ by industry type.

The companies in the different industries of the forestry sector show divergent industrial characteristics such as the market structure, market performance, and product characteristics. Contingency theory insists that the firms' optimal CSR decisions depend on various internal and external factors. Different industries may have different CSR knowledge, motivation, and implementation strategies. Thus, the actual implementation of CSR is considered to be industry-specific [35,47]. Forestry companies in industries with low market concentration and product differentiation may implement more CSR to achieve differentiation strategies and obtain the scarce resources controlled by stakeholders. From the stakeholder view of CSR, different industries have different influential stakeholders whose concerns may be different. Forestry companies in different industries may therefore display different social commitments. Hence, the industry type may affect the CSR of forestry companies.

**Hypothesis 2a (H2a).** *Forestry companies in different industries have different CSR levels.*

**Hypothesis 2b (H2b).** *Forestry companies in different industries implement different CSR strategies.*

2.3.3. Forest Resources and Corporate Social Responsibility

The cultivation and harvesting of forests involve the use of land, indigenous people, and communities. The Chinese government has strengthened the regulations on the use of forestland and the protection of farmers' rights in recent years. Forestry companies with forest resources involve

a wider range of stakeholders and more primary and influential stakeholders than those without forest resources. Based on stakeholder theory, forestry companies with forest resources may be more concerned about social problems and more likely to implement CSR activities than those without forest resources. Their CSR contents and strategies may thus differ.

**Hypothesis 3a (H3a).** *Forestry companies with and without forest resources have different CSR levels.*

**Hypothesis 3b (H3b).** *Forestry companies with and without forest resources have different CSR strategies.*

### 2.3.4. Ownership and Corporate Social Responsibility

The role of ownership in CSR implementation has not yet been conclusively determined. Some scholars believe that different owners have different impacts on a firm's CSR. For example, Xu [52] found that the CSR levels of state-owned companies are higher than those of private and foreign companies. Oh et al. [53] showed a significant positive relationship between CSR rating and ownership by institutions and foreign investors, while shareholding by top managers was negatively associated with a firm's CSR rating. However, Zhang et al. [54] found that the greater the proportion of state-owned shares, the worse is the CSR level. Still, Muller and Kräussl [55] insisted that no significant difference in CSR between state-owned and private companies is expected.

Forestry companies are divided into three categories based on the controlling shareholder: state-owned, private, and foreign. The existing evidence suggests that different owners may have divergent orientations and preferences for strategic decisions. Consequently, CSR implementation should vary depending on the controlling shareholder. The controlling shareholder of Chinese state-owned forestry companies is the state or local government, who aim to maximize all benefits including the social, environmental, and economic benefits. On the contrary, private and foreign forestry companies may not invest heavily in socially responsible activities because the costs of doing so may far outweigh their potential benefits. They are thus more concerned with generating a profit. Hence, less CSR information can be expected in owner-managed and foreign-invested companies [56]. Therefore, ownership may affect the CSR of forestry companies.

**Hypothesis 4a (H4a).** *Ownership affects the CSR levels of forestry companies.*

**Hypothesis 4b (H4b).** *Ownership affects the CSR strategies of forestry companies.*

## 3. Research Methodology

### 3.1. Research Methods

There are no universally accepted CSR scales and there is no consistent understanding of the CSR concept. Therefore, it is difficult to use questionnaires or interviews to study CSR in China. Instead, the present study performed a quantitative content analysis on CSR in Chinese forestry companies. Content analysis is a research technique used to make replicable and valid inferences from texts (or other meaningful materials) in the context of their use [57]. The disclosed CSR information of Chinese listed forestry companies was the object of the content analysis. Chinese listed forestry companies are industrial leaders. Their CSR performances reflect industrial CSR development. Additionally, their CSR disclosures in annual reports, CSR reports, and environmental reports represent their understanding of CSR and what it should constitute. In this study, manual coding was undertaken by two researchers.

The classic procedure of quantitative content analysis involves building the coding table, undertaking the coding process and performing a statistical analysis. With regard to building the coding table, according to the stakeholder theory [58], there are nine categories of stakeholders in forestry companies. Thus, the CSR activities in forestry companies are divided into nine

categories: The environment, employees, communities, general social issues, consumers and products, governments, investors and creditors, supply chains, and other CSR stakeholder activities. The subcategories in previous research have been integrated with these nine categories. After building the preliminary coding table, an exploratory analysis was implemented to adjust the categories and subcategories in the table. The category of other stakeholders' CSR activities and some subcategories such as performance appraisal, community involvement, public health, and establishing a good public relationship with the government were removed from the coding table. Several new subcategories were discovered and added such as environment management system, environment-related training, and clean production. The final coding table includes eight categories and 48 subcategories (Table 1).

**Table 1.** The coding table and levels of corporate social responsibility (CSR) in Chinese forestry companies.

| Category/Subcategory | Level | Proportion (%) | Category/Subcategory | Level | Proportion (%) |
|---|---|---|---|---|---|
| 1 Community | 8 | 100 | 4.4 Employee growth and development | 1 | 0.37 |
| 1.1 Support economic development of the community | 2 | 25 | 4.5 Health and safety of employees | 65 | 23.81 |
| 1.2 Community priority on employment | 2 | 25 | 4.6 Employee training | 54 | 19.78 |
| 1.3 Supporting local culture, education, and sports infrastructure construction | 3 | 37.5 | 4.7 Employee involvement | 11 | 4.03 |
| 1.4 Consider the interests of the community | 1 | 12.5 | 4.8 Employee stock ownership | 14 | 5.17 |
| 2 Consumer and product | 116 | 100 | 4.9 Awards for employees | 4 | 1.47 |
| 2.1 Safety and quality of products | 1 | 0.86 | 5 General social issues | 71 | 100 |
| 2.2 Product maintenance and after-sales service | 3 | 2.59 | 5.1 Consider the interests of vulnerable groups | 13 | 18.31 |
| 2.3 Quick response to clients' requirements | 5 | 4.31 | 5.2 Charity | 58 | 81.69 |
| 2.4 Research and development leaders | 41 | 35.34 | 6 Government | 87 | 100 |
| 2.5 Green marketing | 12 | 10.34 | 6.1 Tax integrity | 8 | 9.20 |
| 2.6 Awards for and certification of products | 48 | 41.38 | 6.2 Support the government | 1 | 1.15 |
| 2.7 Feedback mechanism to customers | 6 | 5.17 | 6.3 Solve employment problems | 2 | 2.30 |
| 3 Environment | 237 | 100 | 6.4 Awards for government | 76 | 87.36 |
| 3.1 Pollution control | 60 | 25.32 | 7 Investor and creditor | 100 | 100 |
| 3.2 Environmental restoration | 3 | 1.27 | 7.1 Guarantee profit | 2 | 2 |
| 3.3 Use of energy and resources | 39 | 16.46 | 7.2 Information disclosure | 26 | 26 |
| 3.4 Recycling | 28 | 11.81 | 7.3 Investor relations management | 13 | 13 |
| 3.5 Environmental products | 10 | 4.22 | 7.4 Communication channel | 30 | 30 |
| 3.6 Green and safe raw materials | 4 | 1.6 | 7.5 Return mechanism | 25 | 25 |
| 3.7 Environmental awards and certification | 34 | 14.35 | 7.6 Maintain shareholders' rights and interests | 4 | 4 |
| 3.8 Environment management system | 25 | 10.55 | 8 Supply chain | 24 | 100 |
| 3.9 Environment-related training | 18 | 7.59 | 8.1 Timely payment | 3 | 12.5 |
| 3.10 Clean production | 16 | 6.75 | 8.2 Integrity and credit | 7 | 29.17 |
| 4 Employee | 273 | 100 | 8.3 Treat all suppliers and agents fairly | 5 | 20.83 |
| 4.1 Sign labor contracts | 15 | 5.49 | 8.4 Provide business partners with support | 3 | 12.5 |
| 4.2 Competitive wages and welfare | 107 | 39.19 | 8.5 Require business partners to conduct social responsibility | 2 | 8.33 |
| 4.3 Relationship with employees | 2 | 0.73 | 8.6 Protect the rights and interests of suppliers | 4 | 16.67 |
| Total | 916 | | | | |

Second, the coding process was implemented. All the CSR information was analyzed under the framework of the coding table. The CSR activities identified from the reports were coded directly under the appropriate subcategories. For example, one subcategory of environmental responsibility is the use of energy and resources. Thus, the related CSR activities were coded as 1, such as energy-saving, environmentally friendly energy, and large-scale projects related to energy conservation. If there was no such activity, it was coded as 0. If the same CSR activity was mentioned many times, it was coded only once. The awards or certifications were also considered to reflect the levels of CSR activities. Therefore, the coding table has special subcategories to reflect the corresponding awards or certifications. Two coders conducted independent coding and, subsequently, the coding results were compared. The consistency coefficient is 0.86, which is higher than the acceptable level ($\geq 0.80$) [59]. Agreement on inconsistent coding was ultimately reached through discussions.

After the data were coded, statistical analyses were conducted. The numbers in each category were treated as the levels of CSR activities. Subsequently, CSR strategies were analyzed from the depth and width perspectives, following Lee and Rhee [41]. The depth and width were measured by the levels and ranges of CSR activities, respectively. The CSR levels were taken as categorical variables, and a K-means cluster analysis was conducted. Subsequently, the CSR strategy types of Chinese forestry companies were obtained. Finally, a statistical analysis was conducted to test the influence of the determinants of CSR in forestry companies. Specifically, this study used Kendall's taub non-parametric correlation analysis to examine the impact of company size on CSR contents, the Kruskal–Wallis H-test to examine the effect of industry and ownership on CSR contents, the Mann–Whitney U-test to examine the impact of forest resources on CSR contents, $3 \times 4$ cross-tabulation to analyze the impact of ownership on the CSR strategy, $5 \times 4$ cross-tabulation to analyze the impact of industry on the CSR strategy, $2 \times 4$ cross-tabulation to analyze the impact of forest resources on the CSR strategy, and the Kruskal–Wallis H-test to analyze the effect of company size on the CSR strategy.

*3.2. Data Resources*

There were 42 forestry companies listed on the Shanghai Stock Exchange and Shenzhen Stock Exchange in China in 2016. Four of these companies were excluded from this study because their main business was not related to forestry or they were specially listed owing to abnormal financial conditions. In abnormal situations, CSR activities may be affected. The final sample included 38 forestry companies. The CSR information came mainly from the Shenzhen (http://www.szse.cn) and Shanghai Stock Exchange websites (http://www.sse.com.cn). The information sources also included the 38 companies' corporate annual reports, 15 CSR reports, and two corporate environmental reports (Table 2).

The levels and strategies were obtained from the content analysis and statistical analyses. The company size variables were taken from each company's financial report. The company size measurement was the natural logarithm of the company's total assets at the end of the year. The ownership variables were determined by the nature of the controlling shareholders. The industry variables were distinguished by the company's main business.

**Table 2.** Names of listed forestry companies and CSR information sources.

| Name of the Listed Forestry Company | CSR Information Source | Name of the Listed Forestry Company | CSR Information Source |
|---|---|---|---|
| Chenming Group | Annual report | Orient Landscape | Annual report |
| Huatai Group | CSR report, annual report | Yuntou Ecology | Annual report |
| Qingshan Paper | CSR report, annual report | Palm Landscape | CSR report, annual report |
| Sun Paper | CSR report, annual report | Yihua Group | Annual report |
| Yueyang Forest and Paper | Annual report | Qumei Home Furnishings Group | Annual report |
| Bohui Paper | CSR report, annual report | Markor Furnishings | CSR report, annual report |
| Guanhao High-tech | CSR report, annual report | Suofeiya | Annual report |
| Hengfeng Paper | CSR report, annual report | Sleemon | Annual report |
| Shandong Jincheng | Annual report | Yotrio | Annual report |
| Jinxing Paper | Annual report, corporate environmental report | Dare Global | CSR report, annual report |
| KAN Special Material | Annual report | Fenglin Group | CSR report, annual report |
| MYS Group | Annual report | Yongan Forestry | CSR report, annual report |
| Minfeng Special Paper | Annual report, corporate environmental report | Tubaobao | Annual report |
| Shanying Paper | Annual report | Shengda Wood | CSR report, annual report |
| Shanghai Green New | Annual report | Guangdong Weihua | Annual report |
| Shixian Paper | Annual report | Jilin Forest Industry Group | CSR report, annual report |
| Zhongshun Group | Annual report | Der Future | CSR report, annual report |
| Xiamen Anne | Annual report | Fujian Jinsen Forestry | Annual report |
| Qifeng New Material | Annual report | Zhongfu Development | CSR report, annual report |

## 4. Results

### 4.1. Contents and Levels of CSR Activities

The content analysis indicated that the CSR activities in the coding table were covered by most of the forestry companies. Eight categories of CSR were identified for the Chinese forestry companies, namely the environment, employees, communities, general social issues, consumers and products, investors and creditors, governments, and supply chains. The results showed that Chinese forestry companies implemented a wide range of CSR activities and were concerned about the interests of a variety of stakeholders.

In total, 916 CSR activities were identified from the disclosed information of the 38 forestry companies. CSR for employees exhibits the highest level of CSR activities (273 out of the 916 CSR activities, or 29.80% of the total), followed by CSR for the environment, consumers and products, investors and creditors, governments, general social issues, supply chains, and community. Among the subcategories of the employee category, competitive wages and welfare (39.19%) ranks first, followed by health and safety of employees (23.81%), and employee training (19.78%). Environmental CSR activities rank second among the eight categories. The top three subcategories in the environmental category are pollution control, use of energy and resources, and environmental awards and certifications. Consumer and product responsibility activities rank third. The top three subcategories are awards and certifications of products (41.38%), research and development leaders (35.34%), and green marketing (10.34%). Some consumer and product responsibility activities that have been implemented by foreign forestry companies did not appear in the CSR disclosure of the Chinese forestry companies, such as fair and orderly market competition and consumer rights protection. There are relatively few subcategories in the product and consumer responsibility category. The investor and creditor responsibility activities

rank fourth among the eight categories. Within this category, the contents and levels of different companies displayed high similarity. The top three subcategories are communication channel (30%), information disclosure (26%), and return mechanism (25%). The least mentioned category is community responsibility, the range of which is relatively narrow. Some community responsibility activities implemented by foreign forestry companies do not appear in the Chinese forestry companies' reports such as community involvement and public health" Tables 1 and 3 show the contents and levels, respectively, of the CSR activities of these companies.

**Table 3.** The proportions of each content category of CSR activities.

| Category | Level | Proportion (%) | Ranking |
|---|---|---|---|
| Communities | 8 | 0.87 | 8 |
| Consumers and products | 116 | 12.66 | 3 |
| Environment | 237 | 25.87 | 2 |
| Employees | 273 | 29.80 | 1 |
| General social issues | 71 | 7.75 | 6 |
| Governments | 87 | 9.50 | 5 |
| Investors and creditors | 100 | 10.92 | 4 |
| Supply chains | 24 | 2.62 | 7 |
| Total | 916 | 100 | |

*4.2. Corporate Social Responsibility Strategies*

Four types of CSR strategies were derived from the cluster analysis. Table 4 shows the mean scores of each CSR category, the numbers of companies belonging to each strategic group, and the significance levels of clustering. The results are significant at the 5% level. The first cluster lags behind all seven other categories with a relatively narrow range and low levels of CSR. This means that the companies in this cluster pay less attention to CSR issues and implement fewer CSR activities. This cluster was termed the reactive CSR strategy, following Lee and Rhee [41]. In the present study, 65.79% of Chinese forestry companies adopted a reactive CSR strategy. This strategy is thus the dominant CSR type among Chinese forestry companies. The second cluster shows the highest level of environmental responsibility, a relatively high level of responsibility for general social issues, and low levels for the other categories. This means that forestry companies in this cluster implement a relatively narrow range of CSR activities and show high levels of specific CSR categories. This cluster was called the focused CSR strategy, following Lee and Rhee [41]. In the current study, 7.89% of Chinese forestry companies adopted a focused CSR strategy. The third cluster performs well in all CSR categories, with higher-than-average levels of CSR in all areas. The companies in this group show the highest levels of CSR for the categories of employees, communities, consumers and products, governments, investors and creditors, and supply chains. This group had a relatively wide range of CSR and exhibited high levels of CSR engagement across categories. This group of forestry companies is more concerned with CSR activities than the other groups. This cluster was termed the proactive CSR strategy, following Lee and Rhee [41]. In the present study, 7.89% of Chinese forestry companies adopted a proactive CSR strategy. The fourth cluster shows relatively high levels of CSR in most CSR categories including employees, communities, general social issues, investors and creditors, and governments. This group had a relatively wide range of CSR and exhibited relatively high levels of CSR engagement across categories. This cluster is called the opportunistic CSR strategy, following Lee and Rhee [41]. In the current study, 18.42% of Chinese forestry companies adopted an opportunistic CSR strategy.

**Table 4.** Final cluster center.

| | Cluster | | | | Sig. |
|---|---|---|---|---|---|
| | **1** | **2** | **3** | **4** | |
| Communities | 0.08 | 0.00 | 0.67 | 0.57 | 0.040 |
| Consumers and products | 1.80 | 3.00 | 11.67 | 3.86 | 0.000 |
| Environment | 3.36 | 23.67 | 12.33 | 6.43 | 0.000 |
| Employees | 3.80 | 2.67 | 22.00 | 14.86 | 0.000 |
| General social issues | 0.24 | 4.00 | 3.67 | 4.71 | 0.003 |
| Governments | 2.08 | .00 | 7.00 | 3.03 | 0.035 |
| Investors and creditors | 2.56 | 1.00 | 4.00 | 3.33 | 0.005 |
| Supply chains | 0.36 | 0.67 | 2.67 | 0.86 | 0.002 |
| Number of cases | 25 | 3 | 3 | 7 | |
| Percentage | 65.79% | 7.89% | 7.89% | 18.42% | |

The average CSR level of each forestry company was calculated as follows. Its CSR level was divided by its CSR range (number of CSR categories). Subsequently, the average CSR level of each forestry company was taken as a proxy indicator of the depth and the CSR range was considered to be an indicator of the width. Finally, the axis was built with the depth and width as the ordinate and abscissa, respectively. Figure 1 shows the distribution of Chinese forestry companies based on the depth and width perspectives.

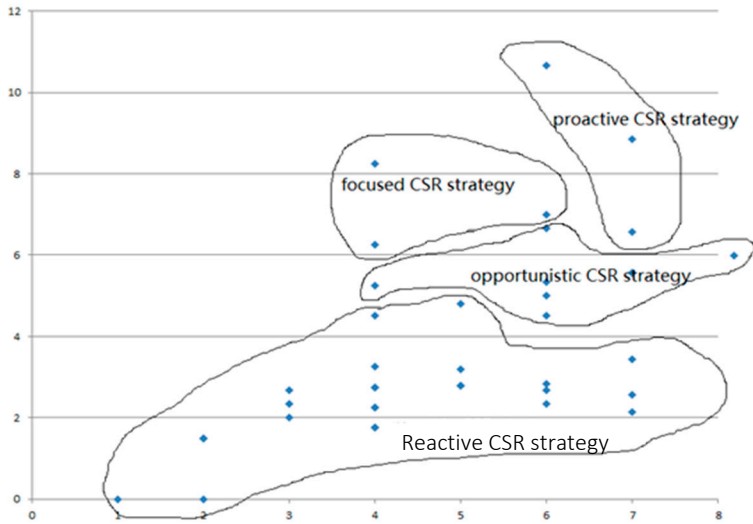

**Figure 1.** The distribution of the CSR strategies of the forestry companies.

*4.3. Determinants of Corporate Social Responsibility in Forestry Companies*

4.3.1. Company Size and Corporate Social Responsibility

The results of the Kendall daub non-parametric correlation analysis indicated a significant correlation between the overall levels of CSR and company size and the levels of environmental responsibility and company size (Table 5). However, significant relationships were not confirmed by the regression analysis between the company size and environmental responsibility levels and the overall levels of CSR (Table 6).

**Table 5.** Correlation test of company size and CSR.

| Category | Sig. | Category | Sig. |
|---|---|---|---|
| Overall levels | 0.034 | Consumers and products | 0.324 |
| Environment | 0.014 | Government | 0.526 |
| Employees | 0.245 | Investors and creditors | 0.530 |
| Communities | 0.636 | Supply chains | 0.969 |
| General social issues | 0.118 | | |

**Table 6.** Regression analysis results.

| Dependent Variable | Independent Variable | Coefficient | Sig. | Adjusted R-Square |
|---|---|---|---|---|
| Overall levels | Company size | 0.310 | 0.059 | 0.071 |
| Environment | Company size | 0.286 | 0.082 | 0.056 |
| Environment | Industry | −0.385 | 0.017 | 0.124 |
| Government | Forest resources | 0.447 | 0.005 | 0.177 |

The Kruskal–Wallis H-test was conducted by taking the CSR strategy as the grouping variable. There is no significant correlation between company size and the CSR strategy ($p = 0.17805$). The choice of CSR strategy is not affected by company size.

### 4.3.2. Industry and Corporate Social Responsibility

A correlation was found between the industry type and corporate environmental responsibility using the Kruskal–Wallis H-test ($p = 0.020$) (Table 7). A significant relationship was confirmed between the industry type and corporate environmental responsibility through the regression analysis ($p = 0.017$). In other words, the industry type influences the levels of corporate environmental responsibility. However, the R-square of the model adjustment is 0.124. This implies that the industry type explains only 12.4% of the variance in the levels of corporate environmental responsibility. The paper industry implements the highest level of corporate environmental responsibility among all industries, while the landscape industry implements the lowest level.

**Table 7.** The significance of the non-parametric tests of industry type and CSR behaviors of forestry companies [a].

| Category | Sig. | Category | Sig. |
|---|---|---|---|
| Overall levels | 0.416 | Consumers and products | 0.399 |
| Environment | 0.020 | Governments | 0.358 |
| Employees | 0.512 | Investors and creditors | 0.074 |
| Communities | 0.409 | Supply chains | 0.950 |
| General social issues | 0.974 | | |

[a] Grouping variable: industry.

The P-value of the Pearson chi-square test of cross-tabulation is 0.678, which indicates that there is no statistically significant correlation between industry type and CSR strategy. The industry type does not influence CSR strategy.

### 4.3.3. Forest Resources and Corporate Social Responsibility

The result of the Mann–Whitney U-test shows that forestry companies with and without forest resources implement different levels of government responsibility ($p = 0.014$) (Table 8). The regression analysis indicates that there is a significant relationship between forest resources and government responsibility ($p = 0.005$). However, forest resources explain only 17.7% of the variance in government responsibility.

**Table 8.** The significance of the non-parametric tests of forest resources and CSR behaviors of forestry companies [a].

| Category | Sig. | Category | Sig. |
|---|---|---|---|
| Overall levels | 0.454 | Consumers and products | 0.194 |
| Environment | 0.658 | Governments | 0.014 |
| Employees | 0.814 | Investors and creditors | 0.822 |
| Communities | 0.085 | Supply chains | 0.507 |
| General social issues | 0.722 | | |

[a] Grouping variable: forest resource.

The chi-square test of cross-tabulation of CSR strategy and forest resources is not significant ($p = 0.086$). Forest resources do not influence the CSR strategies of forestry companies.

### 4.3.4. Ownership and Corporate Social Responsibility

The significance levels of the non-parametric tests of the ownership and CSR behaviors of forestry companies are all greater than 0.05. There is no significant correlation between ownership and CSR levels (Table 9). In other words, there is no difference in CSR levels between state-owned, private, and foreign forestry companies. The cross-tabulation results indicate no statistically significant correlation between the CSR strategy and the ownership of forestry companies ($p = 0.068$). Hence, ownership does not influence the choice of CSR strategy. Table 10 summarizes the results of all the hypothesis.

**Table 9.** The significance of the non-parametric tests of ownership and CSR behaviors of forestry companies [a].

| Category | Sig. | Category | Sig. |
|---|---|---|---|
| Overall levels | 0.258 | Consumers and products | 0.288 |
| Environment | 0.219 | Governments | 0.859 |
| Employees | 0.133 | Investors and creditors | 0.173 |
| Communities | 0.100 | Supply chains | 0.741 |
| General social issues | 0.062 | | |

[a] Grouping variable: ownership.

**Table 10.** Hypotheses and their results.

| Hypothesis | Result |
|---|---|
| **Hypothesis 1a (H1a).** Forestry companies of different sizes exhibit different CSR levels. | Not Confirmed |
| **Hypothesis 1b (H1b).** Forestry companies of different sizes implement different CSR strategies. | Not Confirmed |
| **Hypothesis 2a (H2a).** Forestry companies in different industries have different CSR levels. | Partially Confirmed |
| **Hypothesis 2b (H2b).** Forestry companies in different industries implement different CSR strategies. | Not Confirmed |
| **Hypothesis 3a (H3a).** Forestry companies with and without forest resources have different CSR levels | Partially Confirmed |
| **Hypothesis 3b (H3b).** Forestry companies with and without forest resources have different CSR strategies. | Not Confirmed |
| **Hypothesis 4a (H4a).** Ownership affects the CSR levels of forestry companies. | Not Confirmed |
| **Hypothesis 4b (H4b).** Ownership affects the CSR strategies of forestry companies. | Not Confirmed |

## 5. Discussion

The contents of CSR are diverse in the Chinese forestry sector. CSR activities involve a variety of stakeholders. Chinese forestry companies are attempting to balance economic, environmental, and social considerations. They are implementing similar CSR activities as their international counterparts, albeit with different emphases. Both Chinese forestry companies and their international counterparts focus on environmental and employee responsibility and pay less attention to community responsibility. However, for Chinese forestry companies, employee CSR activities rank first followed by environmental activities. This result is different from the findings on international forestry companies. Some research has concluded that environmental responsibility is the most important in the forestry sector [6,14,35,38]. Employees are one of the most important stakeholders of forestry companies. Most Chinese forestry companies are confronted with the issue of high personnel turnover; thus, they strive to create better work environments and conditions. A joint priority is to attract more skilled or technical workers and increase the retention of employees in Chinese forestry companies. The high proportion of the subcategory, competitive wages and welfare, indicates that forestry companies adopt more economic means to realize employee responsibility such as high salaries and welfare. Non-economic means such as employee development are less utilized. Chinese forestry companies emphasize pollution control, whereas foreign forestry companies emphasize the use of energy and resources. Most foreign forestry companies seek innovations to reduce resource usage and increase energy efficiency [26]. Nowadays, public awareness of environmental protection has intensified and the penalties for the violation of environmental regulations have been enhanced in China. The gap between the proportions of environmental and employee responsibilities is not large.

The economic dimension of CSR activities exceeds the social dimension. This finding is consistent with the previous conclusion that stakeholders have better perceptions of the economic dimension of CSR activities than the social dimension in the Chinese forestry sector [2,7,36]. The economic dimension of CSR activities includes those regarding employees, consumers and products, and investors and creditors. The CSR activities based on the economic dimension have a direct relationship with corporate operations. The social dimension of CSR activities has an indirect relationship with corporate operations, such as CSR activities regarding communities and governments. Chinese forestry companies implement more CSR activities based on the economic dimension that can bring about direct economic benefits and are less involved in CSR activities for which benefits are not directly reflected in revenues. The priority of these companies is profit rather than social and environmental contributions.

This study identified four types of CSR strategies of Chinese forestry companies: Proactive, focused, opportunistic, and reactive. The majority of Chinese forestry companies adopt a reactive CSR strategy. Only 7.89% of forestry companies adopt a proactive CSR strategy. Listed forestry companies are industrial leaders, and it can thus be inferred that CSR strategies for the whole industry should be improved. Despite growing concerns and pressures from consumers, governments, and related organizations in domestic and international markets, the CSR strategies of the Chinese forestry sector are at a relatively inferior level and cannot achieve first-mover advantages. Economic growth is still the main priority of Chinese forestry companies, hence, only a few have implemented a proactive CSR strategy. CSR strategies should improve gradually. From the perspective of a company's dynamic growth, at the beginning of the development phase, the company should adopt a focused CSR strategy and implement more CSR activities associated with the production process that can cut costs and improve revenues, such as reducing pollution, adopting eco-friendly production, saving energy, and improving efficiency. Thereafter, the company can implement more product-oriented CSR activities that can have a more positive influence on investors [60]. Finally, forestry companies could integrate all their CSR activities and develop more proactive CSR strategies. The CSR strategy should correspond to the firm's corporate characteristics. For example, small- and medium-sized forestry companies should choose a focused CSR strategy and should focus limited resources on specific CSR categories. Large companies should choose a proactive CSR strategy to achieve sustainable development and gain competitiveness.

Company size does not influence the CSR levels of forestry companies. From the resource-based view, large forestry companies have easier access to the various resources needed to implement CSR. However, the resources are not the only decisive factor. From the strategy management perspective, small companies are also motivated to implement CSR to achieve differentiation strategies and obtain more access to the resources controlled by influential stakeholders. In other words, both large and small companies are motivated to implement CSR. This result is inconsistent with the conclusion of Han and Hansen [35] because of the different contexts. The understanding of CSR in China is relatively weak, with profits instead of social responsibilities the priority, even if companies have relatively high access to financial resources.

There is a significant difference among industries regarding the levels of corporate environmental responsibility. The paper industry is a pollution-intensive industry. The government, communities, and the media pay more attention to the environmental problems of paper companies than those of other companies in the forestry sector. There are more rules and regulations on the environmental behaviors of paper companies. With greater pressure, paper companies tend to implement activities that reflect greater environmental responsibility. As contingency theory states, the industrial characteristics affect the strategic CSR decision. That is, industries influence the levels of CSR in forestry companies. This result is consistent with the conclusion of Waddock and Graves [48] but differs from the conclusion of Balabanis et al. [51]. This inconsistency may originate from the different samples used. The sample of Balabanis et al. [51] mainly comes from the consumer sector. The difference in industrial characteristics may be relatively weak.

Forest resources partially explain the variance in government responsibility among forestry companies. This is consistent with stakeholder theory. Forestry companies with forest resources have more stakeholders. In particular, the use of forestland and forest harvesting are the targets of direct government supervision. Forestry companies with forestland are more tightly regulated by the government than those without. For example, forest harvesting amounts are assigned by the government. A good relationship with the government therefore affects the operations and the performance of forestry companies. Thus, forestry companies with forestland should pay more attention to government CSR.

Ownership does not influence the CSR levels of forestry companies. Owing to the nature of ownership, it is easier for state-owned forestry companies to obtain resources and government support, which may be scarce for private and foreign forestry companies. Therefore, private and foreign forestry companies are motivated to implement CSR to gain such scarce resources. However, state-owned forestry companies also have social obligations to develop the national economy, maintain social stability, and retain an ecological balance. Thus, it is natural for state-owned forestry companies to implement CSR by, for example, increasing employment, paying taxes, increasing fiscal revenue, and maintaining social stability. Therefore, there is no difference among forestry companies with different ownership. This result is consistent with the conclusion of Muller and Kräussl [55] and different from the opinions of Xu [52], Oh et al. [53], and Zhang et al. [54]. Hence, this issue is controversial and requires more research attention.

## 6. Conclusions and Limitations

### 6.1. Conclusions

The results are as follows. First, the CSR contents of Chinese forestry companies are diverse, including the environment, employees, communities, general social issues, consumers and products, investors and creditors, governments, and supply chains. Second, these companies focus on environmental and employee responsibility and pay less attention to community responsibility. The activities pertaining to employee responsibility are prioritized over environmental activities in these companies. Third, four types of CSR strategies were identified in Chinese forestry companies: Reactive, focused, opportunistic, and proactive. The majority of these companies adopt a reactive CSR

strategy. Only a few choose a proactive CSR strategy. Finally, forest resources partially explain the variance in government responsibility among forestry companies. The industry type influences the CSR levels.

*6.2. Limitations*

This study is based on a quantitative content analysis of the disclosed information of Chinese listed forestry companies. The data sources include corporate annual reports, CSR reports, and environmental reports. It cannot be ruled out that forestry companies still have undisclosed CSR activities. However, the disclosure of CSR information reflects the levels of corporate management. The results are reliable to some extent. The CSR level is represented by the number of CSR activities in Chinese forestry companies, but it does not reflect the intensity of each activity. Some forestry companies may implement a few CSR activities, albeit with a large amount of capital and resources. Listed forestry companies are relatively large in the Chinese forestry sector. Small- and medium-sized forestry companies should be considered in future research to obtain an overall understanding of CSR activities in China's forestry sector.

**Author Contributions:** Conceptualization, Y.L. and L.G.; methodology, Y.L.; software, Y.L.; validation, Y.L. and L.G.; formal analysis, Y.L.; investigation, Y.L.; resources, Y.L.; data curation, Y.L.; writing—original draft preparation, Y.L.; writing—review and editing, L.G.; visualization, Y.L.; supervision, Y.L.; project administration, Y.L.; funding acquisition, Y.L.

**Funding:** This research was funded by Guangdong Natural Science Foundation, grant number 2016A030313406, Foundation for Distinguished Young Talents in Higher Education of Guangdong, grant number 2014WQNCX021, National Natural Science Foundation of China, grant number 71773031 and National Natural Science Foundation of China, grant number 71761147003.

**Conflicts of Interest:** The authors declare no conflicts of interest. The funders had no role in the design of the study; in the collection, analyses, or interpretation of data; in the writing of the manuscript; or in the decision to publish the results.

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
