# Peer review of "Corporate Social Responsibility of Forestry Companies in China: An Analysis of Contents, Levels, Strategies, and Determinants"

_sustainability, doi:10.3390/su11164379_

Round 1

Reviewer 1 Report

The main drawbacks of the paper are the following:

The Abstract must be improved because it have to provide a clear overview of the paper.

The literature review should follow the introduction and be used to justify the objective of the article.

The article does not contain a well-developed and articulated theoretical framework and the logic behind the hypotheses is not persuasive. Thus the hypotheses do not logically flow from the theory.

I suggest to authors to pay attention to the logic behind the hypotheses and to support them by some references.

The methodology is almost unclear and is not adequate for the hypothesis.

There is lack of clear evaluation methods and results. The hypotheses developed for the study were not explicitly defended/refuted. A discussion of the hypotheses in relation to the results is needed.

I suggest to authors to improve the methodology in relationship with the hypothesis.

Good luck!

Author Response

Thank you for your valuable suggestions. All the changes in this manuscript have been marked in blue. The changes are as follows.

First, I have improved the abstract as you suggested. It now covers all the contents that are needed for an abstract.

Second, I have enriched the literature review in order to follow the introduction. 

Third, I offered the theory base for every hypothesis and supported the hypotheses with related literature.

Fourth, I will explain the methodology. This study includes two main purposes: describe the status quo of CSR in the Chinese forestry sector and figure out the determinants of CSR. The methodology for the former is quantitative content analysis. I introduced the process of quantitative content analysis in detail because it was more complicated. The latter is the tests of the hypotheses, which are based on the results of the former part. The methodology for the latter is statistical analyses, which are operated in SPSS. I introduced it briefly because it was easier to understand and operate. I have marked these in purple.

Fifth, I have improved the language.

Finally, the results include two main aspects. One describes the status quo of CSR in the Chinese forestry sector. The other shows the results of the tests of the hypotheses. Corresponding to those, I discuss the two main aspects. The discussion of the hypotheses in relation to the results is marked in purple.

Reviewer 2 Report

I have appreciation for this work- I think the paper will generate significant interest among forest sector scholars. My only suggestions is to broaden the literature it has covered. Specifically, work by Eric Hansen in the context of CSR in the forest products industry and forestry is critically missing.

Author Response

    Thank you for your valuable suggestions. I have read more literature you have recommended and other related literature as well, broadened the literature review, and supported the hypotheses with more references. The added references are as follows.

1.       Han, X.; Hansen, E.; Panwar, R. et al. Connecting market orientation, learning orientation and corporate social responsibility implementation: is innovativeness a mediator?. Scand. J. Forest Res.201328(8), 784-796.http://dx.doi.org/10.1080/02827581.2013.833290

2.       Hansen, E.; Nybakk, E.; Panwar, R. Firm performance, business environment, and outlook for social and environmental responsibility during the economic downturn: Findings and implications from the forest sector. Can. J. Res.2013,43(12), 1137-1144.http://dx.doi.org/10.1139/cjfr-2013-0215

3.       Hansen, E.; Juslin, H. Strategic marketing in the global forest industries.2nd ed. Author's Academic Press:Corvallis, Oregon, 2011.

4.       Panwar, R.; Han, X.; Hansen, E. A demographic examination of societal views regarding corporate social responsibility in the US forest products industry. Forest Policy Econ.201012(2), 121-128.http://dx.doi.org/10.1016/j.forpol.2009.09.003

5.       Vidal, N.G.; Kozak, R.A. Corporate responsibility practices in the forestry sector. J. Corp. Citizenship 200831, 59–75. https://doi.org/10.9774/GLEAF.4700.2008.au.00009

6.       Panwar, R.; Paul, K.; Nybakk, E. et al. The legitimacy of CSR actions of publicly traded companies.J. Bus. Ethics2013125(3), 16.http://dx.doi.org/10.1007/s10551-013-1933-6

7.       Nikolaou, I. A framework to explicate the relationship between CSER and financial performance: An intellectual capital-based approach and knowledge-based view of firm. J. Knowl.Econ.2017. 16.https://doi.org/10.1007/s13132-017-0491-z

8.       Malovics, G.; Csigene, N.N.; Kraus, S. The role of corporate social responsibility in strong sustainabilityJ. Socio- Econ.J. Socio- Econ.200837(3), 907–918. https://doi.org/10.1016/j.socec.2006.12.061

9.       Sodhi, M.S. Conceptualizing social responsibility in operations via stakeholder resource-based view. Prod. Oper. Manag. 201524(9), 1375–1389. https://doi.org/10.1111/poms.12393

10.      Branco, M.C.; Lima Rodrigues, L. Corporate social responsibility and resource-based perspectives. J. Bus. Ethics200669(2):111-132.https://doi.org/10.1007/s10551-006-9071-z

11.      Vidal, N.; Kozak, R.; Cohen, D. Chain of custody certification: An assessment of the North American solid wood sector. Forest Policy Econ. 2005,7(3), 345-355.http://dx.doi.org/10.1016/S1389-9341(03)00071-6

12.      Carroll, A.B. A three-dimensional conceptual model of corporate performance. Acad. Manage. Rev.19794(4), 497–505.http://dx.doi.org/10.5465/amr.1979.4498296

13.      Gordon, G.G. Industry determinants of organizational culture. Acad. Manage. Rev.199116(2), 396–415.http://dx.doi.org/10.2307/258868

14.      Söderholm,, K. Environmental awakening in the Swedish pulp and paper industry: pollution resistance and firm responses in the Early 20th century. Bus.Strategy. Environ.201018(1), 32-42.http://dx.doi.org/10.1002/bse.556

Reviewer 3 Report

It is a well written and interesting paper. Although it is an extremely interesting article, I suggest that some amendments could be made before it is ready for publication. In particular, I suggest authors:

a)      authors should make a little more general description of the introduction and not just only for China. It is better to write down the problems of forest companies and the relationship with CSR and after to present China experience.

b)      In the first paragraph of the literature review, I propose to describe other theories that explain the reasons which firms adopt corporate social responsibility projects such as resource-based theory, knowledge based theory and stakeholder theory. Some indicative references are:

Branco, M. C., & Rodrigues, L. L. (2006). Corporate social responsibility and resource-based perspectives. Journal of business Ethics, 69(2), 111-132.

Sodhi, M. S. (2015). Conceptualizing social responsibility in operations via stakeholder resource‐based view. Production and Operations Management, 24(9), 1375-1389.

Nikolaou, I. E. (2017). A framework to explicate the relationship between CSER and financial performance: An intellectual capital-based approach and knowledge-based view of firm. Journal of the Knowledge Economy, 1-20.

c)      The results do not clearly indicate whether the assumptions made in the theoretical section are confirmed. You may need a table with all the assumptions that show who is confirmed and who does not.

d)      In the discussion, it seems that there is no discussion between the results and international literature in order to show differences with international counterparts as announced in the abstract.

Author Response

Thank you for your valuable suggestions. I have revised my manuscript according to your suggestions. The changes are marked in blue.

First, I have improved the introduction by describing the problems of forestry companies and the relationship with CSR.

Second, I have read the literature you have recommended and other related literature as well, broadened the literature review by describing the reasons for adopting CSR in forest companies, and supported the hypotheses with more references. 

Third, I have created a table (Table 10) that show the results of all hypotheses.

Fourth, this study mainly compared the contents and the priorities of CSR between Chinese forestry companies and their international counterparts. This part is marked in red. We cannot compare the CSR strategies between Chinese forestry companies and their international counterparts because of the absence of related research. 

Round 2

Reviewer 1 Report

The article was improved, but still requires improvements, as following:

The logic behind the hypotheses is not persuasive. Thus the hypotheses do not logically flow from the theory.

I suggest to authors to pay attention to the logic behind the hypotheses and to support them by some references.

There is lack of clear evaluation methods and results. The hypotheses developed for the study were defended/refuted, but not in relationship with the findings.

I suggest to authors to improve the methodology in relationship with the hypothesis.

Good luck!

Author Response

Dear reviewer,

Thank you for your valuable suggestions. I have revised the manuscript according to your review. The changes are as follows:

First, I have improved the hypotheses. I have broadened the literature review and support the hypotheses with persuasive theories. For example, the hypothesis on the relationship between company size and CSR is based on the resource-based theory, stakeholder theory, scale economies theory and the firm visibility perspectives. the hypothesis on the relationship between industry and CSR is based on contingency theory and stakeholder theory. the hypothesis on the relationship between industry and CSR is based on stakeholder theory.

Second, I delete a part of findings which may be confusing and make sure the rest of the finding corresponds to the results.

In addition, my paper includes two parts. The first part is descriptive and the second part is to test the hypotheses. Corresponding to these aims, I use quantitative content analysis and SPSS statistical analysis. I have introduced them in third part including the classic process of quantitative content analysis and SPSS statistical analysis tools which are necessary in this research.

Thank you for your suggestions again.

Best Regard,

Yanli Li

Reviewer 3 Report

Dear authors,

You have successfully answered the  suggested amendments . Thus, I suggest that your  paper is ready for publication in current  form.

Author Response

Dear reviewer,

Thank you for your valuable suggestions. I have learned a lot.

Best Regard,

Yanli Li

Round 3

Reviewer 1 Report

The paper is not sufficiently improved.

Especially, the methodology.

For example, the table 9 and table 10 must be supported by detailed discussions and by results.

The methodology has many lacks and that diminished the relevance of the paper.

I suggest to pay more attention to the relationship between literature review in relationship with the results and discussions.

Good luck!

Author Response

Thank you for your suggestions. I have improved the Results and Discussion sections. The changes are as follows and marked in red in the revised version of the manuscript.

Table 9 shows the results of the non-parametric tests of ownership and CSR in forestry companies, as discussed in detail in the first paragraph of Section 4.3.4. I have added a discussion of the results in Table 9. See paragraphs 7 of the Discussion for more details.

I now explain the results of all hypotheses, compare the results with classical studies, and analyze the reasons for any inconsistencies in paragraphs 4–7 of the Discussion(Section 5).

Table 10 only summarizes the results discussed in Section 4.3 to make it easier for readers to make sense of the findings. All the results in Table 10 have been analyzed in Section 4.3 and discussed in the Discussion (Section 5).

The language has been re-edited by a professional English editor. The language editing certificate is provided in the appendix.